# MYCN Impact on High-Risk Neuroblastoma: From Diagnosis and Prognosis to Targeted Treatment

**DOI:** 10.3390/cancers14184421

**Published:** 2022-09-12

**Authors:** Damiano Bartolucci, Luca Montemurro, Salvatore Raieli, Silvia Lampis, Andrea Pession, Patrizia Hrelia, Roberto Tonelli

**Affiliations:** 1R&D Department, Biogenera SpA, 40064 Bologna, Italy; 2Pediatric Oncology and Hematology Unit, IRCCS, Azienda Ospedaliero-Universitaria di Bologna, 40138 Bologna, Italy; 3Oncodesign SA, CEDEX, 21079 Dijon, France; 4Pediatric Unit, IRCCS, Azienda Ospedaliero-Universitaria di Bologna, 40138 Bologna, Italy; 5Department of Pharmacy and Biotechnology, University of Bologna, 40126 Bologna, Italy

**Keywords:** MYCN, high-risk neuroblastoma, pediatric tumor, neuroblastoma therapeutics, undruggable targets

## Abstract

**Simple Summary:**

Neuroblastoma is one of the most diffuse and the deadliest cancer in children. While many advances have been made in the last few decades to improve patients’ outcome, high-risk neuroblastoma (HR-NB) still shows a very aggressive pattern of development and poor prognosis, with only a 50% chance of 5-year survival. Moreover, while many factors contribute to defining the high-risk condition, MYCN status is well established as the major element in pathology disclosure. The aim of this review is to describe the current knowledge in the diagnosis, prognosis and therapeutic approaches of HR-NB, particularly in relation to MYCN. The review highlights how MYCN influences the HR-NB scenario and the new therapeutic approaches that are currently proposed to target it, in consideration of MYCN as a highly relevant target for HR-NB patient management.

**Abstract:**

Among childhood cancers, neuroblastoma is the most diffuse solid tumor and the deadliest in children. While to date, the pathology has become progressively manageable with a significant increase in 5-year survival for its less aggressive form, high-risk neuroblastoma (HR-NB) remains a major issue with poor outcome and little survivability of patients. The staging system has also been improved to better fit patient needs and to administer therapies in a more focused manner in consideration of pathology features. New and improved therapies have been developed; nevertheless, low efficacy and high toxicity remain a staple feature of current high-risk neuroblastoma treatment. For this reason, more specific procedures are required, and new therapeutic targets are also needed for a precise medicine approach. In this scenario, MYCN is certainly one of the most interesting targets. Indeed, MYCN is one of the most relevant hallmarks of HR-NB, and many studies has been carried out in recent years to discover potent and specific inhibitors to block its activities and any related oncogenic function. N-Myc protein has been considered an undruggable target for a long time. Thus, many new indirect and direct approaches have been discovered and preclinically evaluated for the interaction with MYCN and its pathways; a few of the most promising approaches are nearing clinical application for the investigation in HR-NB.

## 1. Introduction

Neuroblastoma is one of the most diffuse neoplasia in children (<14 years), preceded only by leukemia, lymphomas and central nervous system neoplasms. In 2021, a total of 10,500 children in the USA were diagnosed with cancer, and 6% of these cases were neuroblastoma [1]. Notably, in the last fifteen years the 5-year survival rate has increased from 72% to 81%, highlighting an important advancement in the treatment and therapy of the pathology [1,2], but the problem still remains. In particular, high-risk neuroblastoma (HR-NB) patients, accounting for about half of the overall cases, can be considered more fragile and meaningful due to the poor outcome associated with this condition. In fact, patients affected by HR-NB normally show a strong reduction in 5-year survival with a mean of 50%, in comparison with low and intermediate-risk patients that reach about 90–85% [3]. HR-NB patients normally show mutations in major risk biomarkers such as MYCN amplification [4] and common segmental chromosomal aberration (SCA) [5]. The age of the patient at the pathology diagnosis is considered another important factor, and for this reason a proper evaluation of the tumor is fundamental not only for the treatment of the pathology, but also for better outcomes assessment.

The neuroblastoma risk classification system underwent modification in past decades, from the Evans staging system in 1971 to the most recent International Neuroblastoma Staging System (INSS) or International Neuroblastoma Risk Group Staging System (INRGSS), which was used for the first time in 2005 [6]. Recently, the Children’s Oncology Group revised this system, (Table 1) aiming to improve the correspondence between patient therapy and stage assignation and more highly considering the impact of the risk biomarkers [3]. The re-assignation of patients from INRGSS classes of risk [7] to newly identified classes highlighted that HR patients were well assigned using the previous version of INRGSS (only 3.4% of non-HR shifted to HR), and that MYCN is one of the most relevant factors occurring in a majority of case scenarios in the HR-NB patients [3].

MYCN amplification occurs in almost 25% of all neuroblastoma cases and correlates with HR-NB and poor prognosis [8,9,10]. Interestingly, despite the expectations, MYCN amplification does not always result in a higher expression of mRNA or protein [11,12,13], highlighting that more complex interaction should be considered in HR-NB disclosure [14,15]. While the relevance of the correlation between MYCN amplification and over-expression is still under discussion, the role of N-Myc oncoprotein as a potential target for therapy is well established [16,17,18]. In the same way, the prediction potential of MYCN amplification is considered a standard tool to distinguish tumors subtype and patient prognosis in medical practice [19,20]. For this reason, MYCN emerges as a key component of HR-NB in diagnostic, prognostic and medical procedures, such as in pharmaceutical research.

## 2. Diagnosis of High-Risk Neuroblastoma

A proper diagnostic evaluation can be relevant to perform the best patient assignation to a risk group and to select the best therapy available. The most common and used methods for NB assessment include a combination of histological observation with imaging features and multiple laboratory tests such as fluorescence in situ hybridization (FISH), polymerase chain reaction polymerase chain reaction (PCR), multiplex ligation-dependent probe amplification (MLPA) or array comparative genomic hybridization (aCGH) [4,21,22]. In a similar manner to laboratory tests, imaging also offers a broad spectrum of techniques that have been developed or improved over the years, in order to provide reliable tools to assess neuroblastoma staging and follow-up [23,24].

For example, anatomical imaging techniques such as computed tomography (CT) and magnetic resonance imaging (MRI) represent the diagnostic standard for the precise localization of primary tumor mass and provide anatomical details for consequent loco-regional staging [25,26]. Interestingly, the modern CT-based approach has shown the ability to predict MYCN amplification (MNA) status in neuroblastoma through the generation of radiomics profiles of tumors [27], highlighting the great potential of this technique. In contrast, functional imaging analysis, using positron emission tomography (PET) or the combined approach PET/CT, is less useful when approaching the primary tumor, but it is more reliable for distant metastasis disclosure and proper tumor evaluation after anatomical distortions induced by surgery or radiation [24,28]. Moreover, the use of different tracers in PET, such as ^123^I-MIBG, ^18^F-FDG, and ^99m^Tc-MDP, may allow us to obtain the best imaging performance on neuroblastoma tumors, differing in biological and molecular characteristics and further improving the value of this tool [28].

With the advancement of new technologies, the cellular, molecular, genetic and anatomical features of NB become constantly more fast and accessible for analysis, allowing proper patient assignation [3,29]. In particular, MNA status is considered the strongest indicator for both HR-NB assignation and poor prognosis [29,30,31], and a fast diagnosis can be particularly relevant in consideration of the increase in risk with the age of the patient [3,32,33].

### Circulating Free DNA and Circulating Free Cells

Most of the tests for NB assessment require bioptic material. The biopsy procedure needed for the analysis is an invasive procedure and the tumor mass is not always accessible for recovery and analysis. Moreover, the analysis of a tumor with an abundance of non-malignant cells [34] can be confounding and show MNA heterogeneous pattern results [35,36]. For this reason, a new approach was developed involving the use of circulating free DNA (cfDNA) that was isolated from plasma or serum [37]. Using this so called “liquid biopsy” approach, it is possible to overcome the problem related to the invasiveness of the surgical procedure and the genetic heterogeneity found in solid tissue [38]. The technique is fast and fully reliable for the assessment of MYCN copy number using a PCR based analysis [39,40,41]. Moreover, it was demonstrated that cfDNA can be used in combination with specifically quantitative PCR (q-PCR) to perform an MNA analysis with high sensitivity and specificity in patients with advanced disease [42,43]. 

Interestingly, circulating messenger RNA (mRNA), circulating tumor cells (CTCs) and circulating NB exosomes can also be found in biological fluids and used as biomarkers for diagnosis and prognosis assessment [44]. In particular, CTCs can provide comprehensive tumor profiling involving RNA, protein and/or metabolic information, while cfDNA only contains a genomic statement [45,46]. Of course, this method is more expensive and slower than cfDNA use [47], but it can be extensively considered as complementary in HR-NB evaluation, as suggested in other cancer studies [48,49].

## 3. Current Therapies of High-Risk Neuroblastoma

Current NB therapy in the majority of countries is constituted by three phases: induction, consolidation and maintenance therapies (Figure 1), and lasts approximately 18 months, varying by the patient risk [50,51]. Treatment strategies for each phase may include chemotherapy, surgical resection, high-dose chemotherapy with autologous stem cell rescue, radiation therapy, immunotherapy and isotretinoin, but the modality of the administration of the single procedures mostly depends on patient risk status [52]. Low- and intermediate-risk patients, for example, show high overall survival with minimal therapy approach, involving only surgical resection alone or combined with small chemotherapy administration in the induction phase [53,54,55]. On the other hand, the high-risk group need a more aggressive approach, and each phase is involved in the management of the pathology with the administration of any possible therapy available [52,56,57,58], being more challenging not only for the efficacy assessment, but also for the safety of the patient.

### 3.1. Surgical Resection

Surgical resection is considered the first lane of treatment for neuroblastoma and is mostly relevant in HR patients [59]. In particular, the surgical approach may vary from complete tumor resection to gross tumor resection or biopsy only, and this can have a heavy impact on therapy outcome. Complete or gross tumor resection is not always possible but showed better outcome in comparison to less aggressive strategies such as partial resection or biopsy only in HR patients [60,61,62]. More in detail, tumor resection >90% was specifically associated with better EFS than more partial resection [63]. Unfortunately, surgery cannot be considered an independent procedure and to establish the proper timing for resection in function of chemotherapy is not clear [64,65]. In HR-NB, for example, preoperative chemotherapy is highly suggested for the management of an unresectable tumor [66,67]. More in general, despite many studies reporting that patient treatments with neo-adjuvant chemotherapy prior to surgery may facilitate tumor removal and improve post-surgery outcomes [59,68,69], the impact of surgical timing in primary tumor resection remains controversial and challenging [70]. In particular, defining proper chemotherapy strategies in terms of compound and the number of cycles to administer prior to and post-surgery is still a major issue [65]. While surgical procedure remains fundamental, the location of the primary tumor and the experience of the surgeon, such as the post-operative care of the patients, may impact the results of the therapy outcome. In the same way, amplification of the resection extent may increase both intra-operative and post-operative complication and reduce the therapy compliance or lead to the abandonment of such a life-saving procedure [62].

### 3.2. Multi-Agent Chemotherapy

Multimodal chemotherapies treatment strategies for HR-NB patients may vary from place to place, but they always involve a combination of chemotherapy and both high dose and frequent administration strategies. The most used North American model, for example, involves the use of a five-cycle strategy consisting in the administration of topotecan/cyclophosphamide for cycles 1 and 2, cisplatin/etoposide for cycles 3 and 5, and vincristine/doxorubicin/cyclophosphamide for cycle 4 [51], while the well-established COJEC system (cisplatin [C], vincristine [O], carboplatin [J], etoposide [E] and cyclophosphamide [C]) uses another mix of chemotherapy rapidly administered every 21 or 10 days [71,72]. Of course, the use of chemotherapy at a high concentration and frequency inevitably leads to a broad spectrum of side effects, and although the community effort is focused on increasing patient compliance, long-term toxicity still remains a major issue in HR patients [73]. The most common side effects include growth failure, thyroid dysfunction [74], hearing loss [75,76], ovarian/testicular failure [77], diabetes mellitus, pulmonary dysfunction [73,78], cardiac dysfunction [79,80], renal dysfunction, subsequent malignant neoplasm [81,82] and physiologic impairment [83]. Interestingly, endocrinopathies are one of the most prevalent complications after the treatment of HR-NB using modern therapies [78,84]. This condition is particularly relevant in consideration of the young age of the patient and can strongly impair growth and fertility in the adulthood [85].

### 3.3. Autologous Stem Cell Transplantation

An increase in chemotherapy drug dosage was postulated as a strategy for tumor treatment, highly in consideration of the advancement of support therapy. Preclinical studies on the dose-response of cytotoxic agents demonstrated how it is possible to maintain a linear range, highlighting the possibility of effectively increasing the dose and, thus, the effect [86,87]. Unfortunately, chemotherapy has many side effects and, particularly, myelotoxicity has been evaluated as the most dose-limiting toxicity for chemotherapeutic agents [88]. For this reason, the possibility of rescuing the stem cells from the patient and performing autologous stem cell transplantation (ASCT) after high-dose treatment could highly impact therapy outcome [89,90]. Many studies demonstrated how single [56,91] or more recent tandem transplantation [92] post-chemotherapy can increase event-free survival compared to chemotherapy alone, having a particular impact on HR-NB patients’ outcome. While ASCT remains as one of the most important advancements for HR-NB treatment, the procedure itself is not totally without risk. Infection management [93,94] such as stem cell availability, sorting procedure and processing [95] may be challenging, and advancements are needed to safely improve patient healthcare [89].

### 3.4. Radiation Therapy

Similar to surgery and chemotherapy, radiotherapy (RT) is considered one of the most relevant and common parts of the treatment of HR-NB in the multimodal approach [96,97]. While its function in supporting primary tumor removal is well established [98,99], its role in metastasis management after induction chemotherapy is still uncertain [100,101]. Many advancements were achieved to upgrade this technology, increasing performance both in dose shaping and specificity [102], for example, with the administration of proton therapy [100,103], or ^131^I-MIBG [104,105,106] instead of standard photon RT. Considering the improvement in HR-NB treatment and the increase in patient survival, more attention is paid to radiotherapy’s toxic effect. This effect may include growth and developmental failure, hypothyroidism, gastrointestinal dysfunction, neurocognitive defects, pulmonary and cardiac abnormalities, infertility and secondary cancers. The most frequent side effects are musculoskeletal abnormalities followed by the growth impairment of bones, including scoliosis, kyphosis or short stature [107,108,109], as well as general growth impairment effects [84,110]. Less frequently, sensorineural hearing loss, cardiac dysfunction and secondary malignancies can be observed [84,107], underlining a broad spectrum of toxic effects that must be properly taken into account for patient safety [111].

### 3.5. Anti-GD2 Immunotherapy

Immunotherapy and, in particular, the use of monoclonal antibody in cancer has increased in past years [112,113,114]. Unfortunately, pediatric tumors normally show few tumor-specific antigens, and for this reason immunotherapy is less frequent compared to adult counterparts [115]. Interestingly, neuroblastoma represents an exception; in fact, almost all tumorigenic cells show the expression of aberrant gangliosides [116,117]—such as disialoganglioside GD2—which are nearly absent in the majority of normal tissue [118]. Gangliosides are sialic acid-containing glycolipids [119] that are able to stimulate the immune response, and their use in immunotherapy was demonstrated to improve the outcome in a patient with HR-NB [114,120,121,122]. Dinutuximab, for example, is a commercial chimeric anti-GD2, successfully used for many years for the treatment of HR-NB [123,124,125]. Recently, a new humanized anti-GD2 antibody known as naxitamab has been approved by the Food and Drug Administration for the treatment of neuroblastoma and other GD2-related cancers [126,127]. However, anti-GD2 treatment was also found to be associated with several side effects. Significant neuropathic pains can be found in almost all patients treated with anti-GD2 [128,129], while mydriasis, light accommodation impairment [130] and severe demyelinating polyneuropathy are reported more rarely [131]. Moreover, anaphylactic reaction can occur in association with the development of circulating antibodies [132]. New strategies are in development to overcome this issue [133], but in general, less aggressive approaches appear to be less effective in immune stimulation [134].

### 3.6. Isotretinoin

Isotretinoin (13-cis-retinoic acid; 13-cisRA) is a retinoid that was first approved in 1982 by the Food and Drug Administration for the treatment of severe acne [135]. Despite its native application, isotretinoin has found many other applications, such as maintenance therapy in HR-NB [96,136]. In fact, it was demonstrated that 13-cis-retinoic acid is able to induce both cell differentiation and the arrest of proliferation in neuroblastoma [137,138,139]. Unfortunately, a major issue is represented by the development of tumor resistance by HR-NB relapsed patients [140,141], resulting in this therapy’s great limitation. Cheilitis or dry lips are the major adverse effects found in almost all the patients treated with isotretinoin, but sun sensitivity, xerosis and xerostomia are also very common [142]. While minimum side effects occur, the efficacy of isotretinoin alone is under discussion, and more frequently, combination treatment overcomes its use as a single agent [121,143,144,145].

## 4. MYCN as Prognostic Indicator in High-Risk Neuroblastoma

Despite the advancement in medical standard therapy for the treatment of HR-NB, there is no specific therapy for MNA patients [92,146,147], but this aspect of the pathology becomes increasingly relevant in consideration of its prognostic effect. For example, the prognostic impact of MNA is particularly relevant in infants with stage M disease where both event-free survival (EFS) and overall survival (OS) are higher in an MYCN non-amplified tumor, that shows better outcomes in comparison with MNA patients (EFS and OS: 82.5% and 90.8% versus 36.9% and 44.8%) [3]. In a similar manner, the behavior of an L2 and MS stage tumor is mostly impacted by MYCN status. EFS and OS are lower at any age when MYCN is amplified, while the absence of amplification correlates with better survivability results for the patients [3]. Interestingly, MNA association with worse prognosis appears more pronounced in the context of other favorable prognostic features and can be considered as an indicator for aggressive intervention [148], highlighting how MYCN gene status impacts therapy selection and medical decision making.

While MNA is considered a staple in neuroblastoma diagnosis, as mentioned before, MYCN over-expression has a more controversial role; in a similar manner, MYCN expression has the same prognostic behavior. Some studies reported that MYCN expression is valuable for prognostic purpose, but only in a specific pathology context. In particular, a patient cohort showed that the lowest value in MNA cases has a more negative outcome when MYCN and mRNA levels are higher, in comparison to patients with higher MNA levels where mRNA expression cannot be considered as prognostic [11]. Otherwise N-Myc protein expression shows different prognostic value, being reported as an indicator of poor outcome regardless of MNA status [31]. These findings together suggest that deep investigation on how and when MNA or MYCN expression are predictors of patients’ outcomes needs to be improved, but their value as key elements of prognosis assessment is undeniable, remaining so in the establishment of HR-NB therapy administration.

## 5. MYCN Determines High-Risk Neuroblastoma

Despite MYCN amplification being the first discovered genetic mechanism in neuroblastoma, its role in driving the pathology is not fully understood [149,150,151]. In fact, MYCN amplification leads to deep remodeling of the cancer cell, influencing its apoptosis resistance, its undifferentiated status, its metabolic landscape and immune evasion.

Different studies showed that MYCN over-expression is an obstacle to neuronal differentiation. High-risk and MYCN amplified neuroblastoma present a different transcriptional profile, where different pathways and genes related to differentiation are particularly altered [152]. Indeed, MYCN amplified neuroblastoma cell lines fail to differentiate in response to 13-cis-retinoic acid [8]. The concomitant inhibition of MYCN and the administration of RA is able to reverse this block [152]. Moreover, the ectopic expression of MYCN in precursor cells blocks the differentiation in chromaffin cells.

In addition, MYCN controls both proliferation and apoptosis: many studies showed that over-expression disrupts the cell cycle, leading to maintained apoptosis inhibition and induced proliferation [153]. In fact, blocking MYCN leads to G1 phase cell accumulation and slows down the transition to S phase and PI3K repression (which is known to promote cell growth and proliferation) [154,155,156,157,158]. MYCN also positively affects the expression of other key cellular regulator such as E2 factor (E2F) and inhibitor of differentiation 2 (ID2), which are also involved in cell cycle progression [155,159,160]. As an additional mechanism, MYCN amplification is also associated to TERT expression and telomere anomalies [161,162].

Interestingly, MYCN can promote apoptosis and/or sensitizes cancer cells to cytotoxic drugs [163,164]. MYCN is able to promote the expression of phorbol-12-myristate-13-acetate-induced protein 1 (NOXA), which is a pro-apoptotic regulator. Moreover, E-box elements are present in the promoter of p53, which is the most known onco-suppressor able to stop cell proliferation and induce apoptosis (even if in a significant part of neuroblastoma p53 is found mutated) [164]. As it is known, the murine double minute 2 (MDM2) is a negative regulator of p53 and is over-expressed in different human malignant tumors [165]. In particular, MDM2 is able to reduce p53 levels using the mechanism of binding to p53 with consequent ubiquitination and proteosomal degradation [166,167]. However, MDM2 also promotes the stability of MYCN, while the latter induces MDM2 transcription [168,169,170]. Thus, MYCN can induce the transcription of p53 and MDM2, regulating the balance between proliferation and apoptosis. Over-expression is thought to alter this precarious equilibrium, inducing MDM2 expression and p53 blocking [168,169,170].

Early studies showed how metabolism is deeply altered in cancer cells. In fact, cancer cells are skewed towards rapid ATP production, which is generally obtained through the “Warburg Effect”, where cells rely on glycolysis and mitochondrial respiration is impaired [171,172,173,174]. Cancer cells then use the fatty acids and glutamine as a source for biosynthesis and ultimately sustain the cell growth and proliferation. Moreover, this alteration in the mitochondria leads to reactive oxygen species (ROS) production, while the fatty acid oxidation is used to replenish the NADPH pool in order to prevent excessive oxidative stress [175,176]. Neuroblastoma and, in particular, MYCN amplified tumors are heavily dependent in glutamine, and blocking MYCN leads to the arrest of glutamine transport [177,178]. In addition, MYCN promotes the glycolysis and fatty acid uptake and leads to mitochondria alteration [178,179]. MNA tumors also present different metabolic alterations, leading to an increase in iron uptake. For instance, it has recently been shown that MYCN induces massive lipid peroxidation and cysteine depletion. This leads MNA to be sensible to oxidative stress and especially to ferroptosis [180]. In this context, blocking MYCN in MNA neuroblastoma leads to ROS production (through TRAP1 decrease), which the cancer cell fails to handle, consequently undergoing apoptosis [181]. In addition, MYCN also blocks autophagy and mitophagy (an autophagy sub-pathway used by the cell to recycle damaged mitochondria), and it has been shown in inducible MYCN cell line (TET21N) that the MYCN blocking restores this pathway (OPTN transcription) [181]. All these studies show a strong rewiring of the metabolism by MYCN expression and the fine grain regulation of the redox equilibrium. 

Different studies showed the role of the phosphatidylinositol 3-kinase (PI3K)/mTOR pathway in neuroblastoma [156,182,183]. In fact, the mTOR pathway is known to stabilize N-Myc, and its blocking affects cell growth. Moreover, retinoic acid has been described as being capable of mTOR inhibition. While the N-Myc protein is stabilized by the mTOR complex, it also regulates the expression of different MTOR genes in a positive loop [152,184]. Indeed, MNA cell lines have a higher expression of mTOR genes, and they are more resistant to mTOR inhibitors. Furthermore, it has been shown that mTOR is also negatively associated to the prognosis [152].

MYCN amplification impact is not limited to the cancer cell itself but also to the tumor microenvironment. In fact, MNA cancers remodel the external environment to sustain their growth and the immune evasion [185]. A significant portion of MNA tumors present PD-L1 expression and MHC I complex down-regulation, leading to a suppressive micro-environment [186,187]. In addition, MNAs are also enriched in M2 macrophages and CD4+ T helper 2 cells [185]. Macrophages are also responsible for maintaining a hypoxic environment and lead to the transcription of hypoxia inducible factor (HIF 2α), which ultimately leads to vascularization and metastasis spreading. Indeed, HR-NB that are fast growing are high in the immunostaining for HIF2 [188]. However, neuroblastoma also exploits other strategies such as expressing other immune-suppressive molecules (such as CD276), miRNAs and exosomes release [189]. Overall, both innate and adaptive immune systems seem to be down-regulated [185]. This complex landscape is probably at the origin of the fact that immune-therapy has shown modest results [185,190,191].

MYCN amplification also leads to extracellular matrix (ECM) modification. ECM is often altered in HR-NB with anomalous collagen I deposit and often correlates with bad prognosis [192,193,194,195]. These alterations are also promoted by the hypoxic and inflamed state of the tumor microenvironment [196,197,198]. Collagen I inhibition has shown a promising effect, allowing better chemotherapy delivery [199]. Moreover, different matrix metalloproteinases (MMPs) are altered in neuroblastoma which are linked to bad prognosis, angiogenesis and metastasis promotion [200,201,202].

HIF2 expression, ECM alteration and VEGF expression in HR-NB also lead to new vascularization. These tumors show more aggressive features such as more immature states and more easily spreading metastasis [203,204,205,206,207,208]. Moreover, there is evidence that PI3K kinases promote VEGF expression via MYCN [209,210]. Interestingly, blocking PI3K by SF1126 in neuroblastoma led to reduced MYCN expression, cell death and angiogenesis block, while temporarily increasing the macrophages’ M1 to M2 ratio, showing how all these mechanisms are interconnected [211,212,213,214].

Furthermore, the neuroblastoma tumor microenvironment presents an enrichment of cancer-associated fibroblasts (CAFs) and mesenchymal stromal cells (MSCs). This enrichment correlates with progression and it is important to sustain the tumor growth, micro-vessels formation and progression, showing correlation with poor outcome [215,216,217]. In fact, CAF produces TGFβ (a cytokine with immunosuppressive property) and CCL2, which recruit TAM to neuroblastoma [218,219,220]. In addition, studies have highlighted that the CAF area extension correlates with MYCN amplification [221,222,223]. MSCs are also involved in inducing an immune suppressive environment, leading to the recruitment of ulterior suppressive cells (T regulatory cells and macrophages).

Other than the direct action of MYCN on cellular process leading to neuroblastoma, in the past few years, many epigenetic mechanisms have been discovered regulating MYCN with a specific role in HR-NB development [224]. Micro RNAs (miRNAs) are small single-stranded RNA molecules that function as post-transcriptional RNA regulators [225]. MiRNAs targeting MYCN were found to be particularly important in regulating its expression in neuroblastoma in different ways [8,226]. Some miRNAs targeting MYCN, such as the let-7 family, work as inhibitors [227] and are down-regulated in neuroblastoma, inducing N-Myc protein expression [228,229,230]. These miRNA types are considered the most common, but other miRNAs were found to work with the opposite mechanism. For instance, miRNAs such as the family of miR-17-92 are substantially employed as MYCN up-regulators. Interestingly N-Myc is able to stimulate the expression of miR-17-92 cluster, suggesting the presence of a positive feedback mechanism of regulation between MYCN and the miR-17-92 cluster itself [230,231,232].

Moreover, the natural antisense transcript was found to be able to regulate MYCN expression. In particular, MYCN locus is able to generate an antisense transcript known as MYCNOS (or N-cym) [233]. This transcript originates from the opposite strand of the locus and regulates MYCN as either regulatory long non-coding RNA (lncRNA) or protein. In fact the lncRNA of MYCNOS regulates MYCN promoter through the recruiting of protein in this site [234], while the MYCNOS-encoded protein works as an inhibitor of glycogen synthase kinase 3 β (GSK3beta), stabilizing in this way the NMYC protein [235]. For this reason, high levels of MYCNOS can be found relative to MYCN over-expression and correlate with poor outcome in neuroblastoma [13,236]. Furthermore, MYCNOS itself was found to be regulated from some non-coding RNAs such as lncUSMycN [237] that are able to suppress its expression and indirectly regulate MYCN [234].

Methylation is another regulatory system used by cells to define gene expression and cellular function. While the correlation between MYCN and its methylation status in neuroblastoma is still unclear, many other genes involved in neuroblastoma transformation were well characterized in both MYCN amplified and non-amplified tumors [224,238]. Extensive methylation was found in many onco-suppressor genes with no particular correlation to MYCN status [239]. In particular, miRNAs were reported to be methylated in neuroblastoma cell lines, highlighting their role in tumor progression and poor prognosis [240].

## 6. MYCN as Therapeutic Target

Despite current therapeutic advances and ongoing clinical trials, NB remains a complex medical challenge, especially in the high-risk cases, and the discovery of new therapeutic approaches is needed to improve patient welfare and outcome (Table 2) [241].

In this scenario, MYCN certainly represents an ideal therapeutic target given its correlation with rapid tumor progression, poor prognosis and the limited expression in normal cells and tissue, suggesting high tolerability for an MYCN-specific approach [8,242]. Many attempts and investigations have been made to develop specific inhibitors for N-Myc protein, but both direct or indirect N-Myc modulators (Figure 2) failed to result in an efficient or reliable N-Myc-specific therapy [243]. Unfortunately, N-Myc targeting shows a different issue. Some of these issues can resemble any transcription factor, while some challenges are more peculiar and reside in the lack of specific interaction site on the protein [244,245] or in the homology with the MYC family oncogene, increasing the difficulty of preserving the physiological function of c-Myc protein in normal tissue [153,246]. For this reason, new strategies have been proposed in the hope of overcoming the failure of the precedent attempt to make MYCN a fully available target for HR-NB [8,18,153,247].

### 6.1. BET Inhibitors

Bromodomain and extra-terminal domain family (BET) is a group of epigenetic regulators that consist of four elements, BRD2, BRD3, BRD4 and BRDT [248]. These proteins can bind DNA and recruit P-TEFb complex to start the elongation of the transcription process, activating the RNA pol II and so regulating gene expression [249,250,251]. Interestingly, these factors are required for both MYCN transcription and MYCN-driven transcription, so that their inhibition can have a double effect [153,252,253].

Many BET inhibitors capable of inhibiting MYC or MYCN in vitro were discovered in the past few years [254,255] with a specific function in NB [252,256], but their application with clinical purpose was unsuccessful. However, new BET inhibitors, BMS-986158, BMS-986378 (NCT03936465) and GSK525762 (NCT01587703) are currently in clinical trial phase I with specific application in neuroblastoma (Table 3), maintaining the interest in this class of compound.

### 6.2. HDACs Inhibitors 

It is known that MYCN can modify the genome in many ways [257]. One of these is represented by the ability to induce the transcriptional repression of tumor suppressor genes by N-Myc protein binding to the MIZ1 and SP1 transcriptional activators and resulting in the recruitment of histone deacetylases (HDACs) [258,259,260]. For this reason, HDACs inhibitors are considered a viable route to target MYCN-amplified neuroblastomas [8,261]. These inhibitors can modulate both histone and non-histone proteins inhibiting cancer-related processes, while stimulating the immune response and chemotherapy sensitivity [262,263]. The Food and Drug Administration has approved different HDAC inhibitors such as vorinostat, romidepsin, belinostat and panobinostat, mostly for hematological cancer types such as T-cell lymphoma or multiple myeloma [264,265,266]. In particular, in neuroblastoma, the HDAC-8 appears to correlate with poor prognosis, and many attempts were made to inhibit its function. The specific inhibitors 1-naphthohydroxamic acid (Cpd2) and PCI-34051, for example, showed potential HDAC-8 inhibitory activity and thereby decreased the neuroblastoma cell viability [267]. However, more common HDACs inhibitors such as vorinostat or parabinostat (NCT04897880) are currently in clinical trials for neuroblastoma and have successfully reached clinical phase 2 (Table 3).

### 6.3. PI3K/mTOR Inhibitors

Protein stabilization is fundamental for the correct function of proteins. Affecting this process represents a possibility to deregulate protein function and, therefore, the cellular process. In cerebellar neuron precursor for example, N-Myc protein stability is regulated by Phosphoinositide 3-kinases (PI3K) through AKT and glycogen synthase kinase 3β (GSK3β) [268,269], suggesting its possible role in inhibiting upstream N-Myc signaling. In a similar manner, N-Myc is also indirectly regulated upstream by mammalian target of rapamycin complexes (mTORC), which modulates cell growth and protein synthesis [270,271].

For this reason, several inhibitors interacting with this key element for protein function were developed in the past decade to increase protein degradation and limit their biological effects [272,273]. First generation inhibitors of mammalian target of rapamycin (mTOR), a core protein of mTORC, showed high efficacy in reducing cell viability and N-Myc level in MYCN-amplified NB cells [274]. More interestingly, compounds such as NVP-BEZ235 (dactolisib) and INK128 (sapanisertib) were found to be able to inhibit the activation of entire PI3K/mTOR pathway in specific MYCN tumors, further promoting their role in N-Myc down-regulation [182,275]. Several mTOR inhibitors have already been approved for the therapeutic treatments of different types of cancer [276,277], while in neuroblastoma new clinical trials are ongoing (NCT02337309, NCT03213678) making mTOR inhibition a very promising therapeutic avenue for MYCN-deregulated childhood cancers (Table 3).

### 6.4. Aurora Kinase-A Inhibitors

Aurora kinase A (AURKA) belongs to a family of serine/threonine kinases, named Aurora, mostly involved in cell division process through the regulation of centrosome formation, chromatin condensation and chromosome microtubule interaction [278]. AURKA expression was found to be altered in many cancers, making it a good candidate for therapy [279,280]. In neuroblastoma, N-Myc is able to interact with AURKA, leading to N-Myc stabilization and the limitation of cell cycle arrest in G2/M [281]. On the contrary, AURKA inhibition induces N-Myc degradation and stimulates cell death [282,283]. In combination, the use of AURKA and BRD4 inhibitors can reduce cell viability in a synergistic way in HR-NB cells [284,285], as well as in glioblastoma cells [286]. Despite the well described efficacy of AURKA inhibitors in combination, clinical trials are also available for a single agent. The selective inhibitor LY3295668 [287] for example is actively under examination alone in a clinical phase 1 study (Table 3) in relapse/refractory neuroblastoma (NCT04106219). 

### 6.5. MDM2 Inhibitors

Tumor suppressor p53 is a critical protein for the regulation of apoptosis, cell cycle arrest or DNA damage repair process in response to DNA damage and cellular stress [288,289,290]. This protein has been reported to be mutated in almost 50% of human cancer [291], with related impairment of transcriptional activity [292,293]. In neuroblastoma, TP53 is rarely mutated [294], but it is normally associated with HR-NB and poor outcome [295,296]. The MDM2 oncogene is amplified and/or over-expressed in numerous human malignancies, including neuroblastoma [297,298], showing poor prognosis in this conditions [299]. MDM2 was discovered to be a negative regulator of p53 through a mechanism involving both transcription repression [300] or protein ubiquitination and degradation [301]. Interestingly, MDM2 may interact with N-Myc in a similar manner to p53 in neuroblastoma, resulting in MYCN mRNA stabilization and translation increase [302,303]. For this reason, the p53-MDM2-N-Myc pathway is very interesting as a target for new therapies [17,304]. For example, the MDM2 inhibitor DS-3032b was able to reactivate both in vitro and in vivo TP53 signaling in MYCN-amplified neuroblastoma [296]. A new dual MDM2/MDMX inhibitor ALRN-6924 [305] is under testing (NCT03654716) in a clinical phase 1 trial for neuroblastoma (Table 3), raising new hope for HR-NB treatment using TP53 reactivation.

### 6.6. MYCN Direct Inhibitor

Despite the difficulties in directly targeting the N-Myc protein, advances in chemistry and chemical genomics have created new instruments to overcome this issue in different ways [247,271,306]. Proteolysis targeting chimeras (PROTACs) induce protein degradation by exploiting the ubiquitination mechanism, resulting in viability for undruggable targets [307]. From a structural point of view, PROTACs are heterobifunctional molecules composed of an E3 ubiquitin ligase, covalently linked to another ligand with the ability to recognize the target and to drive the ubiquitination process [307,308]. As small molecules, 10058-F4 and 10074-G5 successfully showed in vitro the ability to bind N-Myc [309] and may be used to develop PROTACs. Further, the use of such molecules provides a new opportunity for screening strategy in MYCN therapeutics [310].

Directly targeting MYCN mRNA or MYCN gene at the level of DNA using MYCN-specific oligonucleotides is another highly promising and valuable approach for the specific, effective and safe treatment of MYCN-related HR-NB and other MYCN-expressing tumors. In recent years, different synthetic oligonucleotides have been developed for the specific silencing of target genes, making this technology always more affordable for both pre-clinical and clinical studies in cancer therapies [311,312]. An in vitro study on neuroblastoma cells with and without MYCN amplification showed that treatment with specific anti-MYCN small interfering RNAs (siRNAs) targeting the MYCN mRNA may cause cell growth arrest, the activation of apoptosis, and differentiation [313]. Moreover, synthetic miRNAs have recently been developed and showed in vitro the ability to stably interact with MYCN mRNA, which is promising for further biological study [314].

An innovative strategy consists in the specific gene expression inhibition at the level of the DNA of the MYCN gene through an antigene peptide nucleic acid (agPNA) oligonucleotide that is specific for MYCN [315]. Blocking the level of transcription by the antigene oligonucleotide strategy has shown pharmacological advantages over the translation block by antisense oligonucleotides. The chemical modification of peptide nucleic acids (PNAs) confer resistance to the degradation of the oligonucleotide by proteasome and nuclease and the ability to potently and specifically target DNA and resulted in relevant pharmacological optimal properties for the antigene strategy [316,317,318]. Antigene oligonucleotide therapy by targeting MYCN transcription has been demonstrated by the novel MYCN-specific agPNA BGA002 [152] in the preclinical treatment of neuroblastoma, and has also great potential in treating other aggressive MYCN-expressing tumors. BGA002 showed higher efficacy compared with MYCN antisense oligonucleotides [152]. BGA002 is able to specifically target a unique sequence on the human MYCN gene [152], resulting in a dose-dependent decrease in MYCN mRNA and protein. This effect causes a potent decrease in viability in a panel of 20 NB cell lines, in a block of different MYCN tumorigenic alterations and in the anti-tumor efficacy of BGA002 in vivo in a MNA NB mouse model [152]. Moreover, the block of MYCN by the anti-MYCN BGA002 is able to reactivate and restore the effectiveness of natural killer immune cells against NB, reverting the role of MYCN as a driver of a tumor immunosuppressive environment which impacts survival in several MYCN-positive tumors [156]. While MNA-NBs are generally resistant to retinoic acid (RA) treatment, the specific inhibition of MYCN expression by BGA002 has been shown to restore the RA response in MNA-NB, leading to a significant increase in survival in an MNA-NB mouse model [123]. The restoration of RA treatment could be beneficial not only for MNA-NB, but also for the treatment of different MNA tumors. BGA002 has been granted orphan drug designation from the European Medicines Agency (orphan registry: EU/3/12/1016) and from the Food and Drug Administration (orphan registry: DRU-2017-6085). Preclinical regulatory safety profile package studies also showed that BGA002 is well-tolerated, and it is now moving to phase I clinical trials in neuroblastoma patients.

Finally, regulating MYCN expression at the DNA amplification level was found to be possible. CRISPR/Cas9 technology is an editing tool that allows the cutting and/or addition of genomic fragments as needed [319]. Studies on cellular and animal models have shown that the CRISPR/Cas9 technique can be effective in treating cancers [320]. For this reason, several clinical trials are underway to evaluate the efficiency of this technology in treating cancers [321,322]. While no treatment is available using this system, a recent study showed that decreasing MYCN copy number by using MYCN-A3 alkylating agent can down-regulate MYCN expression and suppress NB growth in vitro and in a xenograft mouse model [323]. While this approach could be a new method of intervention, safety and toxicity aspects related to the unspecific genomic activities of CRISP/Cas9 technology should be further investigated.

## 7. Challenge

There are many challenges in managing neuroblastoma and especially HR-NB, beginning from diagnosis to prognosis.

### 7.1. Rarity of This Cancer

To address the low number of patients, a number of recent collaborations have sought. These organizations have run clinical trials on an international basis, including the International Society of Pediatric Oncology-Europe Neuroblastoma Association (SIOPEN) [324]; the Children’s Oncology Group (COG) in North America [325]; the European Neuroblastoma Study Group (ENSG) [72]; the German Pediatric Hematology and Oncology Group (GPOH) [7,326]; and the Neuroblastoma Committee of the Japanese Society of Pediatric Oncology (JNBSG) [327].

### 7.2. Diverse Prognosis

The clinical behavior of neuroblastoma is very heterogeneous with cases of spontaneous regression and fatal progression. Treatment is adjusted according to the combination of many prognostic variables, with the intensity of therapy guided by a risk assessment of the projected behavior of the disease. Any prognostic variable that can reliably guide risk group stratification and avoid the potential late effects following unnecessarily aggressive treatment in patients with a more favorable prognosis is highly desirable.

### 7.3. Initial Response Rates Are Not Optimal

The aim of induction therapy is to reduce the primary tumor size to facilitate successful surgery and to diminish the metastatic tumor load burden. A good response to initial induction therapy has been correlated with a better outcome [328], but it is difficult to compare initial response rates for the different response criteria used.

### 7.4. Risk of Relapse

Those high-risk patients that demonstrate a good response to induction and consolidation therapies are still at a significant risk of relapse, and this is due to the presence of minimal residual disease (MRD). The maintenance phase of treatment at the end of therapy aims to eradicate MRD.

### 7.5. Measurement of Disease Extent

The existence of residual disease is predictive of a poor outcome. The standardized operating procedures for detecting MRD by immunocytology using disialoganglioside GD2, and quantitative reverse transcriptase-PCR using tyrosine hydroxylase mRNA published by INRG, should facilitate the comparison of results [329]. Survival after relapse is poor, with no universally effective regime at present. According to the data of the Italian neuroblastoma registry, for stage 4 patients who had progressed or relapsed, the 10-year OS was only 2% [330].

### 7.6. CNS Relapse

Though CNS site in neuroblastoma at diagnosis is rare, it is a site of disease relapse. This could be due to the inability of many chemotherapy agents to cross the blood–brain barrier. Generally, CNS relapse is fatal; however, the outcome could be improved with the early recognition of disease at this site. Moreover, mIBG does not cross the intact blood–brain barrier and so diagnostic and surveillance mIBG imaging can miss CNS disease.

### 7.7. Minimizing Treatment-Related Morbidity

In the group of low- and intermediate-risk neuroblastoma, the aim has been to reduce treatment intensity for reducing toxicity and long-term effects. Neuroblastoma patients without MYCN gene amplification and even those with unresectable disease and no MYCN amplification have had OS rates of 95–100%. However, the treatment of these patients could induce long-term toxicity. The Childhood Cancer Survivor Study (CCSS) calculated the incidence of secondary neoplasms, which was 3.5% at 20 years and 7% at 30 years after diagnosis [331].

### 7.8. Distribution of Age of Patients

Approximately 3–4% of neuroblastoma cases occur in older children, adolescents and young adults [332,333]. Neuroblastoma in this age group has a different biology and clinical course and different responses to therapy [334]. As the number of patients in this group is small, the results of the studies are difficult to compare to the other groups.

### 7.9. Access to New Drugs

The prognosis of high-risk neuroblastoma is poor, so many research groups are searching for innovative therapeutic approaches. However, novel agents are mainly tested within clinical trials, as safety and efficacy data are required by continental and national drug regulatory authorities before the agents can be licensed and made commercially available. Pharmaceutical companies have little incentive for the development of new drugs for low-incidence diseases such as neuroblastoma where, even if a new drug was found to be effective, there would be little regain of the development costs.

## 8. Conclusions

Many advancements have been made in the last few decades from diagnostic to patient management and drug discovery in high-risk neuroblastoma. Improvements in outcome and general patient welfare are undeniable. New diagnostic procedures and our better understanding of biological markers and their role in determining the pathology enhance the incidence of entire treatment improving lifespan and quality of life. Despite all the advancements we have described, effectively targeting high-risk neuroblastoma remains challenging. Many new specific inhibitors are designed to specifically inhibit MYCN expression and its oncological effect, in consideration of its predominant role in defining high-risk designation and poor outcome.

Currently, while many of these inhibitors are in use for different type of tumors, only clinical trials for high-risk neuroblastoma that remains “orphan” from its own therapeutic agent are reported. Medical science needs to make many other efforts to overcome the issue linked to aggressive tumor and target therapy, especially in the field of genes and transcription factors such as MYCN. The availability of a more personalized medicine approach is always more concrete and provides us with new possibilities and insights on treating aggressive tumors such as HR-NB.

## Figures and Tables

**Figure 1 cancers-14-04421-f001:**
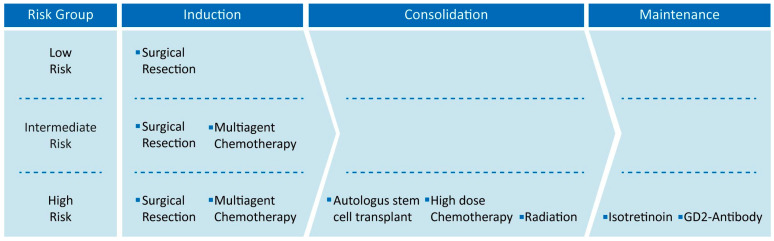
Neuroblastoma therapies are reported in relationship to risk group. Each treatment is collected in its own neuroblastoma therapy phase from induction to maintenance. Low- and intermediate-risk patients only receive the induction phase, while high-risk patients receive all treatments.

**Figure 2 cancers-14-04421-f002:**
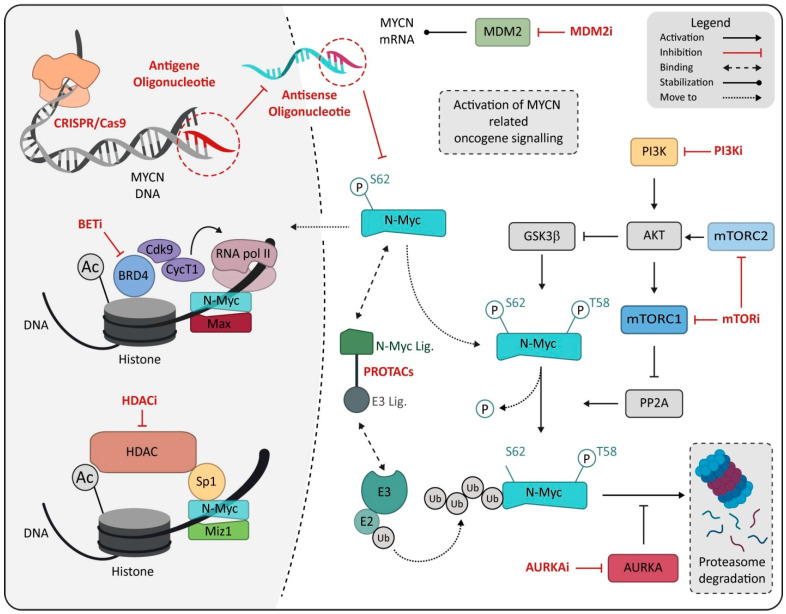
Schematic representation of drugs directly or indirectly targeting MYCN. Drugs are marked in red and reported as class, named on mechanism of action. Abbreviation: MDM2, murine double minute 2; PI3K, Phosphoinositide 3-kinases; PI3Ki, Phosphoinositide 3-kinases inhibitor; BETi, bromodomain and extra-terminal domain family inhibitor; BRD4, bromodomain-containing protein 4; mTORC1, mammalian target of rapamycin complex 1; mTORC2, mammalian target of rapamycin complex 2; mTORi, mammalian target of rapamycin inhibitor; PROTACs, proteolysis-targeting chimeras; MIZ1, MYC-interacting zinc-finger protein 1; HDAC, histone deacetylase; HDACi, histone deacetylase inhibitor; E3, E3 ubiquitin ligase; E2, E2 ubiquitin-conjugating enzyme; AURKA, Aurora kinase A; AURKAi, Aurora kinase A inhibitor; Ub, ubiquitin.

**Table 1 cancers-14-04421-t001:** International Neuroblastoma Risk Group Staging System revised and updated by the Children’s Oncology Group in 2021.

INRGSS	Age	MYCN Amp	SCA at 1 p or 11 q	Ploidy	INPC	Differentiation	Risk Group
L1		No	Any	Any	Any		LR
		AMP					LR or HR
L2	<18 months	No	Absent	DI > 1	FH		IR
			Any	Any	Any		IR
		AMP	Any	Any	Any		HR
	18 months–5 years	No	Any	Any	FH		IR
					UH		HR
	≥5 years	No	Any	Any	UH	Differentiating	IR
						Undifferentiated or poorly differentiated	HR
M	<12 months	No	Any	Any	Any		IR
		AMP					HR
	12 to <18 months	No	Absent	DI > 1	FH		IR
			Present	Any	Any		HR
			Any	DI = 1	Any		HR
				Any	UH		HR
					Any		NA
			At least 1 feature	Unfavorable			HR
		AMP	Any	Any	Any		HR
	≥18 months	Any	Any	Any	Any		HR
MS	<12 months	No bx	No bx	No bx	No bx		LR or IR
		No	Absent	DI > 1	FH		LR or IR
			Present	Any	Any		IR
			Any	DI = 1	Any		IR
				Any	UH		IR
		AMP	Any	Any	Any		HR

Abbreviation: AMP, amplification; bx, biopsy; SCA, segmental chromosome aberration; DI, DNA index; INPC, International Neuroblastoma Pathology Classification; FH, favorable INPC histology; UH unfavorable INPC histology; LR, low-risk; IR, intermediate-risk; HR, high-risk; NA, not applicable. Note: Any included unknown.

**Table 2 cancers-14-04421-t002:** Summary of current therapies and the new therapeutic approach in the treatment of neuroblastoma and high-risk neuroblastoma.

Therapy	Therapeutic Strategy	Availability
Surgical Resection	Tumor mass removal by surgical resection	Standard Medical Practice
Multimodal Chemotherapy	Tumor cell elimination using non-specific chemical agents	Standard Medical Practice
Autologous stem cell transplantation	Stem cell reinfusion after high dose chemotherapy	Standard Medical Practice
Radiation Therapy	Tumor mass removal by radiations	Standard Medical Practice
Anti-GD2 Immunotherapy	Induction of immune system stimulation	Standard Medical Practice
Isotretinoin	Induction of tumor cell differentiation and proliferation arrest	Standard Medical Practice
BET Inhibitors	Inhibition of specific molecular pathway related to MYCN	Clinical Studies
HDACs Inhibitors	Inhibition of specific molecular pathway related to MYCN	Clinical Studies
PI3K/mTOR Inhibitors	Inhibition of specific molecular pathway related to MYCN	Clinical Studies
Aurora Kinase-A Inhibitors	Inhibition of specific molecular pathway related to MYCN	Clinical Studies
MDM2 inhibitors	Inhibition of specific molecular pathway related to MYCN	Clinical Studies
MYCN direct inhibitor	Specific MYCN expression inhibition or N-Myc protein degradation	Preclinical Studies

**Table 3 cancers-14-04421-t003:** Inhibitor drugs under clinical studies for neuroblastoma treatment are reported (ClinicalTrials.Gov, updated 1 August 2022). Drug inhibitor target, clinical phases, trial status and ID are reported.

Drug Name	Target	Clinical Phase	Status	NTC Number
Vorinostat	HDAC	Phase1|Phase2	Recruiting|Completed|No longer available	NCT01019850|NCT03561259|NCT01838187|NCT01132911|NCT02559778|NCT03332667|NCT02035137|NCT01208454|NCT04308330
Panobinostat	HDAC	Phase 2	Terminated	NCT04897880
BMS-986158	BET	Phase 1	Recruiting	NCT03936465
BMS-986378	BET	Phase1	Recruiting	NCT03936465
GSK525762	BET	Phase1	Completed	NCT01587703
SF1126	PI3K/mTOR	Phase 1	Terminated	NCT02337309
Samotolisib	PI3K/mTOR	Phase 2	Recruiting	NCT03213678
ALRN-6924	Dual MDM2/MDMX	Phase 1	Recruiting	NCT03654716
LY3295668	Aurora-A Kinase	Phase 1	Active, no recruiting	NCT04106219
Alisertib (MLN8237)	Aurora-A Kinase	Phase 1|Phase 2	Complete	NCT02444884|NCT01601535|NCT01154816

Abbreviation: HDAC, histone deacetylase; BET, bromodomain and extra-terminal motif; PI3K Phosphoinositide 3-kinases; mTOR, mechanistic target of rapamycin.

## Data Availability

Publicly available datasets at https://www.clinicaltrials.gov/ (accessed on 1 August 2022) were used in this study.

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
