# Peer review of "MYCN Impact on High-Risk Neuroblastoma: From Diagnosis and Prognosis to Targeted Treatment"

_cancers, 2022, doi:10.3390/cancers14184421_

Round 1

Reviewer 1 Report

You provide an overview of the treatment of high risk neuroblastoma as a whole before launching into discussion of potential therapeutic targets in patients with MYCN amplified disease. It would helpful to look at how MYCN amplification impacts prognosis given that therapy is not altered for this biologic finding at this time. The overview of high risk therapy does not seem to be as relevent to the discussion of MYCN amplified disease. This transition from general high risk therapy to MYCN-related targets make the paper seem somewhat disjointed.

Reviewer 2 Report

This review is a collection of the current knowledge in regards to the diagnosis, prognosis and therapeutic approaches of neuroblastoma with a focus on its relation to the N-Myc protein which is often overexpressed in high-risk neuroblastomas, thus being a hallmark for this kind of cancer.

The review is nicely organized. The sections are clear and follow a logical sequence. The information collected is adequate and recent advancements in the field are covered.

Overall the paper is clear and well written but there are multiple grammar mistakes that need to be corrected before being accepted for publication
.

A few other suggestions include the following:

1. The authors should devote a short paragraph in the introduction, presenting some data about the MYCN gene and the N-myc protein and how its overexpression is a hallmark for neuroblastomas. This is transiently mentioned much further in the text.

2. MDM2 is first mentioned in line 251 but there is absolutely no information about it, although it is explained much further in the text (line 451)

3. Section 6 (Challenges) should be presented more briefly. For example there are big parts (e.g. lines 549-561) that do not offer on-point information. Additionally, each challenge should bemarked with a change of paragraph. Right now the section is presented as a huge paragraph that tires the reader.

Reviewer 3 Report

Dear Authors, 

you made a really great work!

The paper is a review on the MYCN impact on high risk neuroblastoma from diagnosis and prognosis to targeted treatment.

The Authors made a great work in terms of methodology and the paper sounds scientific and well written.

However, some improvements are mandatory before acceptance.

The abstract is well written, complete and summary in its various aspects. The keywords are complete and appropriate.

The introduction  is really well written.

In the Diagnosis of High-risk Neuroblastoma:

·       “The most common and used methods for NB assessment include a combination of histological observation with imaging features and multiple laboratory tests such as fluorescence in situ hybridization (FISH), polymerase chain reaction polymerase chain reaction (PCR), multiplex ligation-dependent probe amplification (MLPA) or array comparative genomic hybridization (aCGH) [4,8,9].” I suggest that the authors enrich the imaging aspects of this pathology before moving on to laboratory tests.

In the Current therapies of High Risk Neuroblastoma:

·       I suggest the authors to clarify when Multi-agent Chemotherapy is recommended for Surgical Resection and the link between the two as timing.

Pease check copyright Figure 2.

I suggest the authors to insert a summary of the therapies treated to make the content of the article easily readable.

Conclusions are concise and clear.

Bibliography should be formatted respecting the journal’s requirements and no improper citations are evidenced.

Figures and labels are clear and easy to comprehend.

English is clear and easy to understand.

I just have to congratulate the authors for the good work they have done.

Round 2

Reviewer 1 Report

The additions strengthen this review of MYCN amplification and possible therapeutic targets in HR NBL.   I think the challenges section in particular highlights the problems with advancing care for patients with HR NBL. This paper now warrants publication.